# Ureterovesical Anastomosis Complications in Kidney Transplantation: Definition, Risk Factor Analysis, and Prediction by Quantitative Fluorescence Angiography with Indocyanine Green

**DOI:** 10.3390/jcm11216585

**Published:** 2022-11-07

**Authors:** Andreas L. H. Gerken, Kai Nowak, Alexander Meyer, Maximilian C. Kriegmair, Christel Weiss, Bernhard K. Krämer, Pauline Glossner, Katharina Heller, Ioannis Karampinis, Frank Kunath, Nuh N. Rahbari, Kay Schwenke, Christoph Reissfelder, Werner Lang, Ulrich Rother

**Affiliations:** 1Department of Surgery, University Medical Center Mannheim, Medical Faculty Mannheim, Heidelberg University, Theodor-Kutzer-Ufer 1-3, D-68167 Mannheim, Germany; 2Department of Surgery, RoMed Hospital Rosenheim, Pettenkoferstraße 10, D-83022 Rosenheim, Germany; 3Department of Vascular Surgery, Friedrich Alexander University Erlangen-Nuremberg, Krankenhausstraße 12, D-91054 Erlangen, Germany; 4Department of Urology, University Medical Center Mannheim, Medical Faculty Mannheim, Heidelberg University, Theodor-Kutzer-Ufer 1-3, D-68167 Mannheim, Germany; 5Department of Biometry and Statistics, University Medical Center Mannheim, Medical Faculty Mannheim, Heidelberg University, Theodor-Kutzer-Ufer 1-3, D-68167 Mannheim, Germany; 6Department of Medicine V, University Medical Center Mannheim, Medical Faculty Mannheim, Heidelberg University, Theodor-Kutzer-Ufer 1-3, D-68167 Mannheim, Germany; 7European Center for Angioscience, Medical Faculty Mannheim, Heidelberg University, Ludolf-Krehl-Straße 13-17, D-68167 Mannheim, Germany; 8Department of Nephrology, Friedrich Alexander University Erlangen-Nuremberg, Krankenhausstraße 12, D-91054 Erlangen, Germany; 9Division of Thoracic Surgery, Academic Thoracic Center Mainz, University Medical Center Mainz, Johannes Gutenberg University Mainz, Langenbeckstr. 1, D-55131 Mainz, Germany; 10Department of Urology and Pediatric Urology, Friedrich Alexander University Erlangen-Nuremberg, Krankenhausstraße 12, D-91054 Erlangen, Germany; 11DKFZ Hector Cancer Institute, University Medical Center Mannheim, Medical Faculty Mannheim, Heidelberg University, Theodor-Kutzer-Ufer 1-3, D-68167 Mannheim, Germany

**Keywords:** near-infrared, NIR, renal transplantation, perfusion assessment

## Abstract

Ureteral stenosis and urinary leakage are relevant problems after kidney transplantation. A standardized definition of ureterovesical anastomosis complications after kidney transplantation has not yet been established. This study was designed to demonstrate the predictive power of quantitative indocyanine green (ICG) fluorescence angiography. This bicentric historic cohort study, conducted between November 2015 and December 2019, included 196 kidney transplantations. The associations between quantitative perfusion parameters of near-infrared fluorescence angiography with ICG and the occurrence of different grades of ureterovesical anastomosis complications in the context of donor, recipient, periprocedural, and postoperative characteristics were evaluated. Post-transplant ureterovesical anastomosis complications occurred in 18%. Complications were defined and graded into three categories. They were associated with the time on dialysis (*p* = 0.0025), the type of donation (*p* = 0.0404), and the number of postoperative dialysis sessions (*p* = 0.0173). Median ICG ingress at the proximal ureteral third was 14.00 (5.00–33.00) AU in patients with and 23.50 (4.00–117.00) AU in patients without complications (*p* = 0.0001, cutoff: 16 AU, sensitivity 70%, specificity 70%, AUC = 0.725, *p* = 0.0011). The proposed definition and grading of post-transplant ureterovesical anastomosis complications is intended to enable valid comparisons between studies. ICG Fluorescence angiography allows intraoperative quantitative assessment of ureteral microperfusion during kidney transplantation and is able to predict the incidence of ureterovesical anastomosis complications. Registration number: NCT-02775838.

## 1. Introduction

Kidney transplantation has become the gold standard for the treatment of end-stage renal disease (http://www.transplant-observatory.org/contador1, accessed on 1 November 2022). Most of the surgical transplant procedures are highly standardized and therefore comparable between different transplant centers, leading to relatively constant complication rates [1,2]. However, the anastomosis between the ureter and the bladder is still one of the techniques that continue to evolve. Generally, the incidence of major urological complications like ureteral stenosis or urinary fistula vary between 1 and 13% [3,4]. Different issues have been described to have influence on this [5,6,7].

First, the routine use of ureteric stents: A recent Cochrane review showed that the use of ureteric stents can reduce the complication rates of the ureteric anastomosis from 7–9% to about 1.5% [5,8]. However, this improvement is accompanied by rising infection rates, and bears the risk of stent migration, bladder irritation, and malpositioning.

Second, the surgical technique applied for the implantation of the transplant ureter into the bladder: Different methods have been described including “Taguchi” ureteroneocystostomy and the two most common antireflux techniques: “Politano-Leadbetter” (PL) and “Lich-Gregoir” (LG) [9]. Taguchi ureteroneocystostomy requires a smaller incision of the bladder and is easier to perform in contrast to the antireflux techniques. A systematic review comparing Taguchi to LG concluded that currently there is not enough evidence for one or the other method [10]. Literature comparing the most commonly applied PL and LG techniques mainly demonstrate an advantage for the LG method, as shorter ureters can be used, therefore the risk of ischemic injury is lower [9]. A meta-analysis by Alberts et al. found a lower risk of urinary leakage and hematuria by using this method [11]. Therefore, the LG technique is recommended by the European Association of Urology for kidney transplantation.

Another issue that can lead to complications of the ureteroneocystostomy is malperfusion of the ureter during the organ procurement. However, up to date no standardized method for the visualization of ureter perfusion during the transplant procedure has been implemented.

Near-infrared fluorescence angiography with indocyanine green (ICG) is an emerging tool for intraoperative perfusion control after kidney transplantation. It has been demonstrated that delayed graft function after kidney transplantation can be predicted intraoperatively by applying this technique [12]. A recently published study yielded encouraging results for the detection of early ureteral ischemia [13].

The aim of this study was to identify risk factors for post-transplant ureterovesical anastomosis complications retrospectively on the basis of a new clinical definition of this complication, and to reevaluate the concept of ureteral malperfusion as a risk factor using the intraoperative findings of ICG fluorescence angiography for the assessment of the graft ureteral microperfusion.

## 2. Materials and Methods

### 2.1. Study Design and Patients

A bicentric historic cohort study was conducted including patients who had presented with end-stage renal disease and had underwent kidney transplantation. Deceased-donor and living-donor transplantations were included. All patients listed for kidney transplantation at two university hospitals between November 2015 and December 2019 were screened for study inclusion. In the absence of exclusion criteria, namely known allergy to iodine, ICG, contrast agents, severe hepatic dysfunction, pregnancy, hyperthyroidism, pulmonary hypertension, or allergic diathesis, patients were prospectively enrolled in the study (NCT-02775838) and the kidney allograft and transplant ureter perfusion analysis was conducted intraoperatively, as described below.

A total number of 196 fluorescence videos were suitable for study inclusion into this study (Erlangen, n = 143, and Mannheim, n = 53). The study was conducted in congruence with the Declaration of Helsinki and the Declaration of Istanbul and was approved by the ethics committees of the universities of Erlangen-Nuremberg and Mannheim (162_15B, 2016-513N-MA); all patients gave their written informed consent. The study adhered to the STARD guidelines [14].

### 2.2. Surgical Procedure and ICG Fluorescence Angiography

The standard techniques were used for preoperative diagnostics, organ procurement, and the transplantation procedure. The operative procedures were exclusively performed by surgeons with several years of experience in kidney transplantation. A transurethral catheter was placed into the bladder under sterile conditions. During bench dissection special care was taken to preserve the periureteral fat and blood vessels.

Intraoperative ICG fluorescence angiography of the allograft was performed with the SPY Elite system (STRYKER, Kalamazoo, MI, USA) five minutes after reperfusion. A single bolus of ICG (ICG-Pulsion, Pulsion Medical Systems AG, Munich, Germany, or Verdye, Diagnostic Green, Aschheim, Germany) was administered intravenously at a standardized dose of 0.02 mg/kg body weight [15]. The assessments were performed according to a standardized protocol under comparable circulatory conditions in a shaded operating room to avoid ambient light interference, as described in detail before [12,15,16]. Intraoperative videos were recorded over a period of 138 s to monitor ICG inflow and outflow. To allow subsequent quantification of the perfusion, the kidney allograft and the ureter were both positioned in an area of the camera field at the correct distance to the camera head (25–35 cm), assisted by the built-in laser targeting device. Following fluorescence angiography, the graft ureter was shortened as much as anatomically possible. The results of the perfusion analysis did not influence the intraoperative decision of the extent of ureteral shortening, because the quantification was performed postoperatively (see below). A double-J ureteric stent was inserted into the transplant ureter. The technique of “Lich-Gregoir” was applied for the ureterovesical anastomosis. A surgical drain was placed near the ureterovesical anastomosis. Transurethral catheters were left in place for at least 5 days. The surgical drain was removed after the removal of the transurethral catheter. The double-J ureteric stent was removed 3–4 weeks after transplantation. Kidney allograft ultrasonography was performed in every patient repeatedly in the first days after transplantation during postoperative surveillance on the IMC unit and at least one time before discharge to exclude allograft hydronephrosis. Furthermore, it was performed after removal of the ureteric stent, and in any patient with postoperative micturition problems.

### 2.3. Analysis of Fluorescence Angiography Video Sequences and the Ureteral Perfusion

The fluorescence angiography videos in a gray scale of 256 different shades were analyzed postoperatively regarding the change of fluorescence intensity over the time using the integrated SPY-Q analysis tool-kit (SPY-Q, Stryker). Four parameters of microperfusion are defined in the quantitative analysis by the SPY-Q software: “Ingress”, “Ingress Rate”, “Egress”, and “Egress Rate”. The analysis regarding cortical microperfusion of the kidney allograft and quantification has been described in prior studies [15,16]. Quantitative assessment of graft ureter microperfusion was performed in analogy to that. The ingress rate quantifies the inflow in terms of the increase of the fluorescence intensity per second (increase in gray stats per second). The egress rate is a parameter of the outflow of blood, measured as the decrease in fluorescence intensity per second. Ingress represents the difference between the initial baseline fluorescence intensity and the maximum intensity assessed, and egress is the difference between maximum intensity and final intensity. Microperfusion was evaluated separately for upper pole, middle part, and lower pole of the kidney and for each third of the ureter (proximal/medial/distal) by selecting regions of interest (ROI) as displayed in Figure 1.

### 2.4. Grading of Post-Transplant Ureterovesical Anastomosis Complications

The periprocedural transplant as well as patients characteristics were assessed. The ureter function was monitored up to one year after kidney transplantation. Therefore, all ureter revision procedures after allograft transplantation were recorded. According to clinical considerations, the type of ureteral complication was graded into three different categories (Grade A–C) as shown in Table 1. A normal postoperative course (no ureteral complication) was attested, if the intraoperatively inserted double-J stent was removed within 4 weeks after transplantation without sonographic signs of allograft hydronephrosis.

### 2.5. Statistical Analysis

Assuming a complication rate of about 20%, a power analysis provided that the sample size of 196 patients is sufficient to detect a medium effect regarding a quantitative variable with a power of 0.90.

All statistical calculations were performed using SAS statistical software, release 9.4 (SAS Institute Inc., Cary, NC, USA). Quantitative variables are presented as mean values together with standard deviations or (e.g., in the case of skewed variables) as median values together with minimum and maximum. For qualitative factors, absolute and relative frequencies (percentages) are given. The comparison of two independent groups (e.g., complications versus no complications) was performed using the Fisher’s exact test for qualitative factors, the Cochran–Armitage trend test for ordinally scaled parameters, or the Mann–Whitney U test for quantitative variables, respectively. In order to compare more than two groups, the Kruskal–Wallis test was applied. For the comparison of the ureter thirds regarding ingress or ingress rate, the Friedman test was performed supplemented by Wilcoxon tests for two paired samples in case of a significant test result.

In order to investigate the correlation between two quantitative variables of perfusion assessment, the Spearman correlation coefficient was assessed.

Furthermore, logistic regression analyses were performed to identify parameters potentially associated with ureterovesical anastomosis complications. For each regression model, the AUC (area under the ROC curve) was calculated as a measure of goodness.

In general, *p* < 0.05 was considered to show a statistically significant test result.

## 3. Results

### 3.1. Patients and Procedure Characteristics

In total, 196 patients were included in this historic cohort study (141 deceased-donor and 55 living-donor kidney transplantations). Mean recipient age was 54 (±14) years. The study included 132 male (67%) and 64 female (33%) patients. The median time on dialysis was 40 (0–171) months.

Regarding vascularization, 142 kidney grafts were transplanted with a single-artery supply, 45 grafts had 2 and 4 grafts had 3 arteries. In 17 grafts there was a separate upper pole and in 12 grafts there was a separate lower pole artery. The majority of organs was drained by a single vein. Eight organs had two and one organ had three veins. All of the kidneys had a single ureter. The median operating time was 172 (64–433) minutes. The median cold ischemia time was 566 (0–1680) minutes and the median warm ischemia time was 28 (12–121) minutes.

### 3.2. Incidence of Ureterovesical Anastomosis Complications

Post-transplant ureterovesical anastomosis complications occurred in 36 patients (18%). Most of the patients with ureterovesical anastomosis complications were classified as grade A (22 patients, 61%), i.e., they could be treated conservatively by prolonged retrograde ureteric stenting. A complication grade B, that required nonsurgical antegrade endourologic intervention, occurred in 2 patients (6%) and a complication grade C requiring invasive surgical treatment developed in 12 patients (33%).

### 3.3. Association between Recipient, Donor, and Periprocedural Characteristics and Ureterovesical Anastomosis Complications

The subgroups of patients with any grade of ureterovesical anastomosis complication (grade A–C) and without complications (grade 0) are compared in Table 2. Three parameters were found to have a significant influence on the occurrence of post-transplant ureterovesical anastomosis complications: The number of months on dialysis (*p* = 0.0025), the type of donation (*p* = 0.0404), and the number of postoperative dialysis sessions (*p* = 0.0173). The length of cold ischemia time was not significantly associated with the occurrence of overall complications (*p* = 0.0991).

### 3.4. Intraoperative Graft Ureteral Perfusion Analysis

A representative example of quantitative intraoperative ureteral microperfusion during kidney transplantation is presented in Figure 2a. Regarding all patients, median ICG ingress at the proximal third of the ureter was 21.5 (4–117) AU, at the middle third it was 17 (2–107) AU and at the distal third it was 13 (0–100) AU. Median ingress rate was 2.7 (0.1–23.8) AU at the proximal, 1.9 (0–17.1) AU at the middle, and 1.2 (0–13.4) AU at the distal third of the ureter. Friedman yielded significant differences of ICG ingress and ingress rate between the three different localizations (each *p* < 0.0001). Pairwise comparison between the thirds also yielded significant differences (ingress: proximal vs. middle *p* = 0.0027, middle vs. distal *p* < 0.0001, proximal vs. distal *p* < 0.0001; ingress rate: each *p* < 0.0001).

### 3.5. Intraoperative Kidney Allograft Perfusion Analysis

Median ICG ingress of the renal allografts was 156.00 (8.00–253.00) AU, ingress rate was 28.90 (0.10–78.90) AU. Regarding the two poles of the kidney, ingress at the upper pole was 130 (5.00–254.00) AU, at the lower pole was 125.00 (2.00–254.00) AU. Ingress rate at the upper pole was 23.10 (0–89.00) AU, and at the lower pole it was 21.00 (0–81.10) AU.

### 3.6. Correlation of Ureteral Perfusion with Kidney Perfusion

ICG ingress and ingress rate at the proximal ureter correlated significantly positive with the equivalent perfusion parameters of the whole kidney and the three different parts of the kidney (Table 3). Notably, the correlations of ureteral and kidney perfusion parameters were stronger at the lower pole of the kidney (ingress: r = 0.47387, ingress rate: r = 0.54978) as compared to the upper pole (ingress: r = 0.35483, ingress rate: r = 0.33189, each *p* < 0.0001).

### 3.7. Association between Intraoperative Perfusion Analysis and Ureterovesical Anastomosis Complications

Both parameters of ureteral and kidney perfusion assessment characterizing inflow (ingress, ingress rate) showed significant differences between the subgroup of patients with any grade of ureterovesical anastomosis complication (grade A–C) and without complications (grade 0), as presented in Table 4.

The outflow parameters (egress, egress rate) generally show larger ranges and they did not always differ significantly between both groups, particularly regarding the middle and distal parts of the ureter. An example of a pattern of reduced intraoperative ureteral microperfusion in a patient with postoperative complications is shown in Figure 2b. Specific threshold values of the ICG perfusion parameters ingress and ingress rate in AU at the proximal ureteral third and the lower pole of the kidney as predictors for post-transplant ureterovesical anastomosis complications are presented in Table 5 together with values for sensitivity and specificity.

### 3.8. Perfusion Parameters in Different Grades of Ureterovesical Anastomosis Complications

The median ingress at the proximal ureter in Grade A ureterovesical anastomosis complications was 14.00 (5.00–25.00) AU. In Grade B it was 21.50 (10.00–33.00) AU and in grade C it was 14.00 (9.00–29.00) AU. There was no significant difference between the groups regarding ingress (*p* = 0.8387) as well as ingress rate (*p* = 0.7035).

Comparing the perfusion at the proximal ureter in grade C ureterovesical anastomosis complications with the remaining patients (i.e., grades 0–B), a significant difference was detected for median ingress (14 (9.00–29.00) AU vs. 22 (4.00–117.00) AU, *p* = 0.0243), but not for ingress rate (2.00 (0.20–4.50) vs. 3.05 (0.10–23.80), *p* = 0.0945). At the middle and distal third of the ureter, there was no significant difference for ingress or for ingress rate.

### 3.9. Risk Factors for Grade C Ureterovesical Anastomosis Complications

Univariable logistic regression analysis revealed an association of the following parameters with grade C post-transplant ureterovesical anastomosis complications: months on dialysis (*p* = 0.0016), operation time (*p* = 0.0144), URp ingress (*p* = 0.0243), cold ischemia time (*p* = 0.0409), URp ingress rate (*p* = 0.0945), URp egress (*p* = 0.0939), URm ingress (*p* = 0.0763), and URd ingress (*p* = 0.0628).

Multiple logistic regression analysis (starting with a model containing the parameters cold ischemia time, URp ingress and URp ingress rate, using backward selection method) revealed only cold ischemia time being significantly associated with grade C complications. The ROC analysis of the parameter “cold ischemia time” yielded an optimum cutoff value of 792 min, with a sensitivity of 67% and a specificity of 77% (AUC = 0.704, *p* = 0.0256) for the prediction of grade C complications.

## 4. Discussion

Urological complications occur frequently after renal transplantation. In the recently published literature, the incidences range between 3.6% and 15.5% [4,18,19,20,21,22,23,24], but rates up to 30% have been reported [3,25]. Their occurrence in immunosuppressed patients is associated with relevant morbidity, including negative implications for long-term graft survival [26,27], and even with an increased mortality rate [4]. The results of this study confirm ureteral malperfusion and the duration of cold ischemia as being the key risk factors for post-transplant ureterovesical anastomosis complications after kidney transplantation. Intraoperative fluorescence angiography with ICG can be implemented in the future as a tool for individual risk stratification.

Owing to the lack of a standard definition of ureterovesical anastomosis complications after renal transplantation, it is difficult to compare our results with those from prior studies. This is also reflected in the broad variation in incidence rates reported in different trials. The two most common subtypes of ureterovesical anastomosis complications are stenosis/stricture/obstruction and urinary leakage/fistula [25]. Stenosis occurs in about 3% [4,18,25,26] and leakage/fistula in 1–4% [4,20,24,25,28]. Management strategies are similar for all subtypes and include mainly ureteric stenting, but also surgical revision [29,30,31]. Existing definitions suggest that unspecific laboratory findings, such as elevated serum creatinine [32], any event requiring percutaneous nephrostomy or surgical revision [33], or the combination of symptomatic events with the need for intervention [27] are indicators for relevant ureteral complications or they classify the anatomic site of lesion [34]. However, the majority of publications do not even mention a definition. To the best of our knowledge, this is the first study suggesting a standardized clinical definition of postrenal transplantation ureterovesical anastomosis complications. In analogy to the existing definitions of lymphatic complications after kidney transplantation [35], we define three grades of post-transplant ureterovesical anastomosis complications, as presented in Table 1. The severity grading is based on the invasiveness of the management strategies, in particular the necessity of a prolonged period of ureteric stent placement (grade A), the need for antegrade (endo)urologic interventions including percutaneous nephrostomy (grade B), and the need for reoperations (grade C). Applying our proposed definition, the overall rate of ureterovesical anastomosis complications of 18% is rather high in comparison to the previous studies mentioned above because our definition registers all patients, even those with a slightly abnormal postoperative course (grade A). The rate of severe complications (grade C) was 6% in our study, which is comparable with the outcomes of other studies that mention the rate of surgical revisions [20,33].

Besides technical errors, one of the main pathogenic mechanisms for all subtypes of complications is believed to be ureteral malperfusion [4,25] leading to either segmental ureteral stenosis, or urinary leakage due to necrosis or compromised anastomotic healing. However, it is has been technically difficult to measure ureteric perfusion objectively in order to provide direct evidence for this theory. Fluorescence angiography is an emerging tool for this purpose that fulfills the requirements of allowing real-time, in situ, noninvasive, quantitative tissue perfusion measurements without exposing the patients to the relevant side effects. A recently published study comparing the results of intraoperative ureteral ICG perfusion assessment with histopathological findings of 31 sections of dissected ureters showed that ICG fluorescence angiography had a sensitivity of 100% and a specificity of 93% for the detection of ureteral ischemia [13]. According to the authors, optimal perfusion was only maintained in the proximal 14 cm of the ureters. Our results of intraoperative ureteral and allograft perfusion imaging confirm that the ICG fluorescence signal decreases significantly with the length of the ureter. Furthermore, we showed that the ICG ingress at the proximal ureter exhibits predictive power for ureterovesical anastomosis complications. For the ureteral perfusion analysis with ICG, only the inflow parameters (ingress, ingress rate) seem to be representative. A possible explanation for this observation could be that venous drainage at this early point in time (five minutes after reperfusion) is not fully established, particularly at the distal parts of the ureter. We observed a correlation of the perfusion intensity of the lower kidney pole with the ureter. The perfusion of the lower kidney pole was an independent predictor for ureterovesical anastomosis complications. These observations can be explained by considering ureteral vascularization. Ureteric blood supply depends on branches of the renal artery [36], particularly the lower polar artery branches that traverse in periureteric tissue [4]. In consequence, extensive periureteric dissection should be avoided to preserve the arterial blood supply during harvesting and bench dissection. Although ureteral length alone could not explain ureteral complications in previous studies [36,37], our results suggest that graft ureters should be shortened as much as possible while at the same time allowing tension-free anastomosis to the bladder.

In the present study, the parameters time since dialysis initiation, type of donation, and need for postoperative dialysis were associated with post-transplant ureteral complications. As shown before, spending a long time on dialysis can lead to bladder atrophy, which increases the risk for ureterovesical anastomosis complications [18,38]. Further risk factors for general postoperative urological complications after kidney transplantation that have been identified in large retrospective case series include male recipient gender [21,26,27,33], technical aspects [1,4,27,28], cold ischemia time [18], extended criteria donors [39], donor age, and delayed graft function [40]. In contrast to other studies [33,41,42], arterial multiplicity did not influence the development of ureterovesical anastomosis complications in our patient cohort, but this is most likely due to the limited number of patients expressing this feature here. In the present study, cold ischemia time was associated with severe ureterovesical anastomosis complications (grade C).

The regular use of quantitative intraoperative fluorescence angiography during kidney transplantation can potentially impact clinical practice. The technique can be employed as an objective intraoperative quality control of kidney allograft and ureteral microperfusion before and after completion of the ureterovesical anastomosis. This procedure would allow fluorescence-guided interventive decision-making on the optimal length of the ureter before performing the anastomosis. ICG angiography can be performed repeatedly allowing further shortening or reduction of tension after completing the anastomosis in case of limited perfusion. However, the potential of intraoperative decision-making was not evaluated in this study, but our findings support the performance of further prospective studies using the standardized endpoint definition provided here. In the present study, we focused on the predictive value of the proximal ureteral microperfusion for postoperative anastomosis complications. Our results can be used for cases in which the transplant ureter cannot be shortened further during the operation due to anatomical reasons. In these cases an individual risk prediction can be performed on the basis of the intraoperative perfusion assessment with the aim to facilitate postoperative management: In patients with a decreased intraoperative ICG ingress at the proximal ureter and/or the manifestation of further risk factors, prolonged ureteric stenting could possibly be beneficial in terms of preventing the manifestation of urinary leaks or stenosis, especially considering that outpatient ureteric stent removal is a procedure which can be performed safely [43].

The retrospective design limits the validity of this study. Nevertheless, its design is adequate to support the introduction of a definition and also for risk analysis. Furthermore, due to the retrospective setup and the focus on perfusion assessment, some potentially relevant risk factors for ureterovesical anastomosis complications could not be assessed here, such as the impact of the surgeon’s experience (which was homogeneous), the surgical technique for ureteral implant (only the LG method was applied), and the influence of the preoperative bladder volume capacity (not assessed consistently). Further studies are recommended to provide prospective validation of the definition and assess the influence of all potential risk factors. The ROC analysis performed here showed a good AUC and significant *p*-value that prove the predictive power of ICG ingress for postoperative complications. These findings have clinical relevance because they can be used for intraoperative early individual risk assessment to guide subsequent postoperative management. The sensitivity and specificity of threshold values, however, were only 70%. Thus, they should be validated in further prospective trials with the aim of changing the postoperative management of ureteral complications. ICG fluorescence angiography is a method that is limited by the depth of penetration of near-infrared light. Ureteral grafts are ideally procured with a layer of surrounding fatty tissue. This tissue can sometimes make fluorescence imaging difficult. It is possible that this phenomenon influenced the quantification findings. Furthermore, a small degree of anastomotic tension would probably negatively affect the ureteral microcirculation. The SPY-Q quantification analysis could also be performed intraoperatively for quality control and to allow intraoperative decision-making. In the present study, however, no second perfusion assessment was conducted before completion of anastomosis because the evaluation of ICG fluorescence as a tool for intraoperative decision-making was beyond the scope of this trial, but it could be implemented in further trials.

## 5. Conclusions

The proposed standardized definition of post-transplant ureterovesical anastomosis complications should unify the reporting of this complication while enabling comparison with the results from different studies. Moreover, this study demonstrates that ICG fluorescence angiography can safely be applied for objective intraoperative documentation of the quality of ureteral microperfusion during kidney transplantation. This is the first study providing threshold values for ICG ingress at the proximal ureteral third and the lower pole of the kidney for predicting ureterovesical anastomosis complications. These findings can be used for intraoperative quality control, for decision-making, and for individual postoperative risk stratification potentially improving the postoperative management of patients after kidney transplantation.

## Figures and Tables

**Figure 1 jcm-11-06585-f001:**
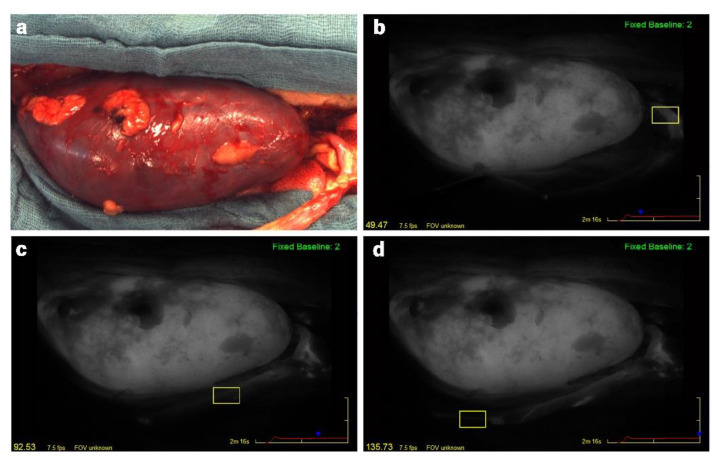
Intraoperative photo (**a**) and quantitative assessment of the graft ureter microperfusion (**b**–**d**) during kidney transplantation with ICG fluorescence analysis (SPY Elite). The ureter is positioned in the right bottom. Microperfusion was evaluated separately for each third of the ureter with the SPY-Q software by selecting regions of interest (yellow boxes): (**b**) proximal; (**c**) middle; (**d**) distal ureteral third.

**Figure 2 jcm-11-06585-f002:**
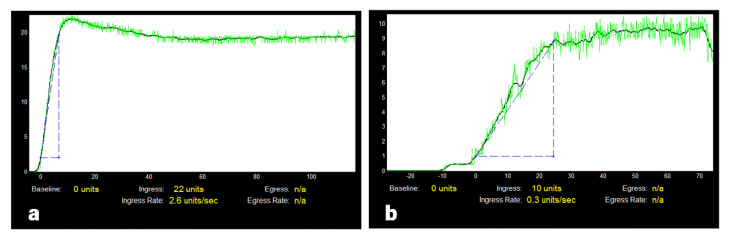
Intraoperative quantitative ureteral microperfusion at the proximal third in ICG fluorescence angiography during kidney transplantation: (**a**,**b**) show examples of output files after quantification with SPY-Q. The graphs illustrate the increase and decrease of ICG fluorescence intensity over a period of 138 s after systemic administration of ICG: (**a**) typical ureteral perfusion pattern, (**b**) reduced perfusion in a patient with a posttransplant ureterovesical anastomosis complication.

**Table 1 jcm-11-06585-t001:** Definition and severity grading of postoperative ureterovesical anastomosis complications after kidney transplantation.

Postrenal transplantation ureterovesical anastomosis complications: Postoperative urinary leakage or stenosis of the ureterovesical anastomosis which requires (endo)urological interventions or surgical revision after kidney transplantation. Urinary leakage: Persistent secretion of urine (drainage output >50 mL/24 h and drain/plasma creatinine >2) [17] from the surgically inserted drains or, after drainage removal, postoperative fluid accumulation (urinoma) related to ureterovesical anastomosis, or extravasation of contrast agent during retrograde pyeloureterography. Ureteral stenosis: Narrowing of the ureter detected directly in retrograde pyeloureterography or indirectly by signs of allograft hydronephrosis in sonography or any other imaging modality.	
**Grade**	**Definition**
Grade A	Temporary ureteral stenosis or leakage after 4 weeks postrenal transplantation that can be managed conservatively, e.g., by prolonged or repetitive retrograde ureteric stent placement.
Grade B	Stenosis or leakage requiring antegrade (endo)urological intervention(s) including percutaneous nephrostomy, or permanent ureteric stent.
Grade C	Ureteral stenosis, necrosis, or leakage requiring reoperation.
Grade 0	No complication (i.e., intraoperatively inserted double-J stent is removed within 4 weeks after transplantation without sonographic signs of allograft hydronephrosis).

**Table 2 jcm-11-06585-t002:** Comparison of recipient, donor, and periprocedural characteristics between recipients with the occurrence of postoperative ureterovesical anastomosis complications and those without kidney transplantation (n = 196). Quantitative variables are expressed as median, minimum, and maximum. For qualitative factors, absolute and relative frequencies are given.

	Complications (°A–C) n = 36	No Complication (°0) n = 160	*p*-Value
Recipient characteristics			
Age (years)	59 (27–74)	55 (6–76)	0.1956
Gender (♀; ♂)	14 (39%); 22 (61%)	50 (31%); 110 (69%)	0.3772
Body mass index (kg/m^2^)	25 (18–39)	26 (13–36)	0.1070
Months on dialysis	88 (7–162)	31 (0–171)	0.0025
Diabetes	10 (29%)	24 (15%)	0.0836
Hypertension	32 (91%)	142 (90%)	1.0000
Donor characteristics			
Age (years)	57 (22–83)	59 (8–83)	0.9616
Gender (♀; ♂)	17 (57%); 13 (43%)	76 (51%); 73 (49%)	0.6895
Procurement and periprocedural characteristics			
Deceased donor; living donor	31 (86%); 5 (14%)	110 (69%); 50 (31%)	0.0404
Arterial supply (1/2/3 arteries)	30 (86%)/ 5(14%)/0	112 (72%)/40 (26%)/4 (3%)	0.0734
Pole artery (upper; lower)	2 (6%); 2 (6%)	15 (11%); 10 (8%)	0.7992
Venous outflow (1/2/3 veins)	35 (100%)/0/0	147 (94%)/8 (5%)/1 (1%)	0.1621
Operating time (minutes)	187 (101–408)	169 (64–433)	0.1289
Cold ischemia time (minutes)	661 (71–1680)	566 (0–1431)	0.0991
Warm ischemia time (minutes)	29 (14–92)	28 (14–121)	0.4261
No. postoperative dialysis sessions	2 (0–20)	0 (0–20)	0.0173

**Table 3 jcm-11-06585-t003:** Comparison of recipient, donor, and periprocedural characteristics between recipients with the occurrence of postoperative ureterovesical anastomosis complications and those without after kidney transplantation (n = 196). Quantitative variables are expressed as median, minimum, and maximum. For qualitative factors, absolute and relative frequencies are given.

	KD Upper Pole	KD Middle Part	KD Lower Pole	KD Total
URp ingress	0.35483	0.40619	0.47387	0.47800
URp ingress rate	0.33189	0.54641	0.54978	0.55497

The table contains Spearman correlation coefficients (r values). Legend: KD, kidney; UR, ureter; URp, proximal third of ureter.

**Table 4 jcm-11-06585-t004:** Association between intraoperative ureteral and kidney perfusion assessment with ICG fluorescence angiography and the occurrence of postoperative ureterovesical anastomosis complications after kidney transplantation.

Perfusion Parameter	Complications (°A–C) Median AU (Range)	No Complication (°0) Median AU (Range)	*p*-Value
URp ingress	14.00 (5.00–33.00)	23.50 (4.00–117.00)	0.0001
URp ingress rate	1.70 (0.10–4.90)	3.35 (0.10–23.80)	0.0007
URp egress	7.00 (3.00–14.00)	12.00 (2.00–88.00)	0.0042
URp egress rate	0.60 (0.10–5.30)	0.90 (0.00–10.20)	0.2263
URm ingress	13.00 (2.00–51.00)	19.00 (4.00–107.00)	0.0236
URm ingress rate	0.90 (0.10–10.00)	2.20 (0.00–17.10)	0.0256
URm egress	6.00 (1.00–42.00)	8.00 (2.00–56.00)	0.3501
URm egress rate	0.60 (0.10–4.00)	1.00 (0.00–9.20)	0.3741
URd ingress	10.00 (0.00–56.00)	14.50 (2.00–100.00)	0.0196
URd ingress rate	0.40 (0.00–12.80)	1.40 (0.10–13.40)	0.0105
URd egress	4.00 (2.00–46.00)	7.00 (0.00–66.00)	0.1713
URd egress rate	0.30 (0.00–4.90)	0.80 (0.00–19.50)	0.1619
KD ingress	129.00 (8.00–252.00)	173.00 (8.00–253.00)	0.0030
KD ingress rate	20.00 (0.10–49.50)	31.50 (0.20–78.90)	0.0004
KD egress	77.00 (1.00–175.00)	102.00 (2.00–200.00)	0.0075
KD egress rate	6.50 (0.10–33.40)	13.35 (0.10–39.70)	0.0072
KDl ingress	73.00 (2.00–254.00)	133.00 (8.00–254.00)	0.0002
KDl ingress rate	12.80 (0.00–54.80)	24.70 (0.10–81.10)	0.0002
KDl egress	59.00 (11.00–138.00)	85.00 (4.00–214.00)	0.0014
KDl egress rate	4.15 (0.30–19.80)	11.60 (0.30–53.00)	0.0008

Legend: KD, kidney; KDl, lower kidney pole; UR, ureter; URd, distal third of ureter; URm, middle third of ureter; URp, proximal third of ureter.

**Table 5 jcm-11-06585-t005:** Cutoff values of the ICG perfusion parameters ingress and ingress rate given in AU at different localizations for post-transplant ureterovesical anastomosis complications.

Perfusion Parameter	AUC	Cutoff (AU)	Sensitivity (%)	Specificity (%)	*p*-Value
URp ingress	0.725	16	70	70	0.0011
URp ingress rate	0.702	2.70	86	58	0.0023
KDl ingress	0.699	125	81	55	0.0004

Legend: AU, arbitrary units; KDl, lower kidney pole; URp, proximal third of ureter.

## Data Availability

The data presented in this study are available on request from the corresponding author. The data are not publicly available due to ethical restrictions.

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
