# Peer review of "Ureterovesical Anastomosis Complications in Kidney Transplantation: Definition, Risk Factor Analysis, and Prediction by Quantitative Fluorescence Angiography with Indocyanine Green"

_jcm, 2022, doi:10.3390/jcm11216585_

Round 1
Reviewer 1 Report
The article entitled “Ureterovesical anastomosis complications in kidney transplantation: Definition, risk factor analysis, and prediction by quantitative fluorescence angiography with indocyanine green” evaluated the associations between quantitative perfusion parameters of near-infrared fluorescence angiography with indocyanine green and the occurrence of ureterovesical anastomosis complications. These complications occurred in 18% of the patients who were submitted to a kidney transplant, and was associated with time on dialysis, the type of donation and the number of postoperative dialysis sessions. The study is interesting and contributes to knowledge in the field. Some technical questions need to be addressed.
1) As the time on dialysis was a risk factor for ureteral complication and a crucial aspect is bladder size (atrophy), why a more robust surgical technique for ureteral implant, such as Leadbetter Politano or ureter-ureter anastomosis, was not performed?
2) What was the bladder volume capacity before kidney transplant?
3) What is the impact of the surgeon´s experience (time of surgical practice, number of transplants, etc?) on the ureteral complications? As living kidney transplant usually takes place during the day (it is scheduled in advance), as opposed to deceased kidney transplant (can take place during night), was this aspect also considered a risk factor for surgical complications?
4) Did the author anticipate the regular use of quantitative fluorescence angiography with indocyanine green or for which group of kidney transplant recipients would this approach be most beneficial?
5) How often was kidney allograft ultrasonography performed in the post-transplant period to detect allograft hydronephrosis?
6) How the current findings would change the post-transplant management?
Reviewer 2 Report
the authors presented an interesting study on the complications rate of ureterovesical anastomosis in kidney transplantation by using the fluorescence angiography. The study is well conducted but there are some points that should be clarified: why the authors preferred to perform the fluorescence angiography immediately after reperfusion? While this is undoubtedly an useful tool for the evaluation of graft and ureter perfusion, it is less useful, in my opinion, in the evaluation of ureterovesical anastomosis, since many other factors may hamper ureter perfusion after anastomosis. So, to quantify the ureter perfusion, it should be better to perform late analysis. But the most important question is: how the fluorescence angiography may impact the clinical practice? Have you modified your surgical technique basing on the fluorescence analysis? Did you discard some grafts basing on fluorescence analysis? This is worth of a comment, to evaluate the real clinical usefulness of fluorescence in this setting.
Reviewer 3 Report
Thank you for inviting me to review this well-written paper. The authors presented an innovative method to quantitate the blood perfusion of the ureter and predict the complication of ureterovesical anastomosis. It will be a great contribution to the field. I suggest accepting it for publication.
Author Response
Response to Reviewer 3 Comments
Point 1: Thank you for inviting me to review this well-written paper. The authors presented an innovative method to quantitate the blood perfusion of the ureter and predict the complication of ureterovesical anastomosis. It will be a great contribution to the field. I suggest accepting it for publication.
Response 1: Thank you very much for your positive review.
Round 2
Reviewer 2 Report
The authors addressed all suggestions